# Limits of Private Learning with Access to Public Data

**Noga Alon**
Department of Mathematics
Princeton University
nalon@math.princeton.edu

**Raef Bassily**[*]
Department of Computer Science & Engineering
The Ohio State University
bassily.1@osu.edu

**Shay Moran**
Google AI
Princeton
shaymoran1@gmail.com

## Abstract

We consider learning problems where the training set consists of two types of examples: private and public. The goal is to design a learning algorithm that satisfies differential privacy only with respect to the private examples. This setting interpolates between private learning (where all examples are private) and classical learning (where all examples are public).

We study the limits of learning in this setting in terms of private and public sample complexities. We show that any hypothesis class of VC-dimension $d$ can be agnostically learned up to an excess error of $\alpha$ using only (roughly) $d/\alpha$ public examples and $d/\alpha^2$ private labeled examples. This result holds even when the public examples are unlabeled. This gives a quadratic improvement over the standard $d/\alpha^2$ upper bound on the public sample complexity (where private examples can be ignored altogether if the public examples are labeled). Furthermore, we give a nearly matching lower bound, which we prove via a generic reduction from this setting to the one of private learning without public data.

## 1 Introduction

In this work, we study a relaxed notion of differentially private (DP) supervised learning which was introduced by Beimel et al. in [BNS13], where it was coined *semi-private learning*. In this setting, the learning algorithm takes as input a training set that is comprised of two parts: (i) a private sample that contains personal and sensitive information, and (ii) a "public" sample that poses no privacy concerns. We assume that the private sample is always labeled, while the public sample can be either labeled or unlabeled. The algorithm is required to satisfy DP only with respect to the private sample. The goal is to design algorithms that can exploit as little public data as possible to achieve non-trivial gains in accuracy (or, equivalently savings in sample complexity) over standard DP learning algorithms, while still providing strong privacy guarantees for the private dataset. Similar settings have been studied before in literature (see "Related Work" section below).

There are several motivations for studying this problem. First, in practical scenarios, it is often not hard to collect reasonable amount of public data from users or organizations. For example, in the language of consumer privacy, there is considerable amount of data collected from the so-called "opt-in" users, who voluntarily offer or sell their data to companies or organizations. Such data is deemed by its original owner to have no threat to personal privacy. There are also a variety of other

---

[*]Part of this work was done while visiting the Simons Institute for the Theory of Computing.

sources of public data that can be harnessed. Moreover, in many scenarios, it is often much easier to collect unlabeled than labeled data.

Another motivation emerges from several pessimistic results in DP learning that either limit or eliminate the possibility of differentially private learning, even for elementary problems such as one-dimensional thresholds which are trivially learnable without privacy constraints [BNSV15, ALMM19]. It is therefore natural to explore whether a small amount of public data circumvents these impossibility results.

A third motivation arises from the following observation: consider a learning problem in which the marginal distribution $\mathcal{D}_\mathcal{X}$ over the domain $\mathcal{X}$ is completely known to the algorithm, but the target concept $c : \mathcal{X} \to \{0, 1\}$ is unknown. One can show that in this setting every VC class can be learned privately with (roughly) the same sample complexity as in the standard, non-private, case. The other extreme is the standard PAC-setting in which both $\mathcal{D}_\mathcal{X}$ and $c$ are unknown to the algorithm. As mentioned earlier, in this case even very simple classes such as one-dimensional thresholds can not be learned privately. In the setting considered in this work, the distribution $\mathcal{D}_\mathcal{X}$ is unknown but the learner has access to some public examples from it. This naturally interpolates between these two extremes: the case when $\mathcal{D}_\mathcal{X}$ is unknown that corresponds to having no public examples, and the case when $\mathcal{D}_\mathcal{X}$ is known that corresponds to having an unbounded amount of public examples. It is therefore natural to study the intermediate behaviour as the number of public examples grows from 0 to $\infty$. The same question can be also asked in the "easier" case where the public examples are labeled.

We will generally refer to the setting described above as *semi-private learning*, and to algorithms in that setting as *semi-private learners*. (See Section 2, for precise definitions.) Following previous works in private learning, we consider two types of semi-private learners: those that satisfy the notion of *pure* DP (the stronger notion of DP), as well as those that satisfy *approximate* DP. We will call the former type *pure* semi-private learners, and call the latter *approximate* semi-private learners.

**Main Results**

In this work we concentrate on the sample complexity of semi-private learners in the agnostic setting. We especially focus on the minimal number of public examples with which it is possible to learn every VC class.

1. **Upper bound:** Every hypothesis class $\mathcal{H}$ can be learned up to excess error $\alpha$ by a *pure semi-private* algorithm whose *private* sample complexity is (roughly) $\mathsf{VC}(\mathcal{H})/\alpha^2$ and *public* sample complexity is (roughly) $\mathsf{VC}(\mathcal{H})/\alpha$. Moreover, the input public sample can be *unlabeled*.

   Recall that $\mathsf{VC}(\mathcal{H})/\alpha^2$ examples are necessary to learn in the agnostic setting (even without privacy constraints); therefore, this result establishes a quadratic saving.

2. **Lower bound:** Assume $\mathcal{H}$ has an infinite *Littlestone dimension*[2]. Then, any *approximate* semi-private learner for $\mathcal{H}$ must have *public* sample complexity $\Omega(1/\alpha)$, where $\alpha$ is the excess error. This holds even when the public sample is *labeled*.

   One example of a class with an infinite Littlestone dimension is the class of thresholds over $\mathbb{R}$. This class has VC dimension 1, and therefore demonstrates that the upper and lower bounds above nearly match.

3. **Dichotomy for pure semi-private learning:** Every hypothesis class $\mathcal{H}$ satisfies *exactly* one of the following:

   (i) $\mathcal{H}$ is learnable by a *pure* DP algorithm, and therefore can be semi-privately learned without any public examples.

   (ii) Any *pure* semi-private learner for $\mathcal{H}$ must have public sample complexity $\Omega(1/\alpha)$, where $\alpha$ is the excess error.

**Techniques**

**Upper bound:** The idea of the construction for the upper bound is to use the (unlabeled) public data to construct a finite class $\mathcal{H}'$ that forms a "good approximation" of the original class $\mathcal{H}$, then

reduce the problem to DP learning of a finite class. Such approximation is captured via the notion of $\alpha$-covering (Definition 2.7). By standard uniform-convergence arguments, it is not hard to see that (roughly) $\mathsf{VC}(\mathcal{H})/\alpha^2$ public examples suffice to construct such an approximation. We show that the number of public examples can be reduced to only about $\mathsf{VC}(\mathcal{H})/\alpha$, even in the agnostic setting. Our construction is essentially the same as a construction due to Beimel et al. [BNS13], but our proof technique is different (see the "Related Work" section for a more detailed comparison).

**Lower bounds:** The lower bounds boil down to a *public-data-reduction lemma* which shows that if we are given a semi-private learner whose public sample complexity is $<< 1/\alpha$, we can transform it to a *fully* private learner (which uses no public examples) whose excess error is a small constant (say $1/100$). Stated contra-positively, this implies that if a class can not be privately learned up to an excess loss of $1/100$ then it can not be semi-privately learned with $<< 1/\alpha$ public examples. This allows us to exploit known lower bounds for private learning to derive a lower bound on the public sample complexity.

**Related Work:** Our algorithm for the upper bound is essentially the same as a construction due to Beimel et al. [BNS13]. Although [BNS13] focuses on the realizable case of semi-private learning, their analysis can be extended to the agnostic case to yield a similar upper bound to the one we present here. However, the proof technique we give here is different from theirs. In particular, our proof relies on and emphasizes the use of $\alpha$-coverings, which provides a direct argument for both the realizable and agnostic case. We believe the notion of $\alpha$-covering can be a useful tool in the analysis of other differentially private algorithms even outside the learning context.

There are also several other works that considered similar problems. A similar notion known as "label-private learning" was considered in [CH11] (see also references therein) and in [BNS13]. In this notion, only the labels in the training set are considered private. This notion is weaker than semi-private learning. In particular, any semi-private learner can be easily transformed into a label-private learner. Another line of work considers the problem of private knowledge transfer [HCB16], [PAE+17], [PSM+18], and [BTT18]. In this problem, first a DP classification algorithm with input private sample is used to provide labels for an unlabeled public dataset. Then, the result is used to train a non-private learner. [BTT18] gives sample complexity bounds in the setting when the DP algorithm is required to label the public data in an online fashion. Their bounds are thus not comparable to ours.

## 2 Preliminaries

Let $\mathcal{X}$ denote an arbitrary domain, let $\mathcal{Z} = \mathcal{X} \times \{0,1\}$ denote the examples domain, and let $\mathcal{Z}^* = \cup_{n=1}^\infty \mathcal{Z}^n$. A function $h : \mathcal{X} \to \{0,1\}$ is called a concept/hypothesis, a set of hypotheses $\mathcal{H} \subseteq \{0,1\}^{\mathcal{X}}$ is called a concept/hypothesis class. The VC dimension of $\mathcal{H}$ is denoted by $\mathsf{VC}(\mathcal{H})$. We use $\mathcal{D}$ to denote a distribution over $\mathcal{Z}$, and $\mathcal{D}_\mathcal{X}$ to denote the marginal distribution over $\mathcal{X}$. We use $S \sim \mathcal{D}^n$ to denote a sample/dataset $S = \{(x_1, y_1), \ldots, (x_n, y_n)\}$ of $n$ i.i.d. draws from $\mathcal{D}$.

**Expected error:** The expected/population error of a hypothesis $h : \mathcal{X} \to \{0,1\}$ with respect to a distribution $\mathcal{D}$ over $\mathcal{Z}$ is defined by $\mathsf{err}(h; \mathcal{D}) \triangleq \mathbb{E}_{(x,y)\sim\mathcal{D}} [\mathbf{1}(h(x) \neq y)]$.

A distribution $\mathcal{D}$ is called *realizable by* $\mathcal{H}$ if there exists $h^* \in \mathcal{H}$ such that $\mathsf{err}(h^*; \mathcal{D}) = 0$. In this case, the data distribution $\mathcal{D}$ is described by a distribution $\mathcal{D}_\mathcal{X}$ over $\mathcal{X}$ and a hypothesis $h^* \in \mathcal{H}$. For realizable distributions, the expected error of a hypothesis $h$ will be denoted by $\mathsf{err}(h; (\mathcal{D}_\mathcal{X}, h^*)) \triangleq \mathbb{E}_{x\sim\mathcal{D}_\mathcal{X}} [\mathbf{1}(h(x) \neq h^*(x))]$.

**Empirical error:** The empirical error of a hypothesis $h : \mathcal{X} \to \{0,1\}$ with respect to a labeled dataset $S = \{(x_1, y_1), \ldots, (x_n, y_n)\}$ will be denoted by $\widehat{\mathsf{err}}(h; S) \triangleq \frac{1}{n} \sum_{i=1}^n \mathbf{1}(h(x_i) \neq y_i)$.

**Expected disagreement:** The expected disagreement between a pair of hypotheses $h_1$ and $h_2$ with respect to a distribution $\mathcal{D}_\mathcal{X}$ over $\mathcal{X}$ is defined as $\mathsf{dis}(h_1, h_2; \mathcal{D}_\mathcal{X}) \triangleq \mathbb{E}_{x\sim\mathcal{D}_\mathcal{X}} [\mathbf{1}(h_1(x) \neq h_2(x))]$.

**Empirical disagreement:** The empirical disagreement between a pair of hypotheses $h_1$ and $h_2$ w.r.t. an unlabeled dataset $T = \{x_1, \ldots, x_n\}$ is defined as $\widehat{\mathsf{dis}}\,(h_1, h_2; T) = \frac{1}{n} \sum_{i=1}^n \mathbf{1}\,(h_1(x_i) \neq h_2(x_i))$.

**Definition 2.1** (Differential Privacy [DMNS06, DKM$^+$06]). *Let $\epsilon, \delta > 0$. A (randomized) algorithm $\mathcal{A}$ with input domain $\mathcal{Z}^*$ and output range $\mathcal{R}$ is called $(\epsilon, \delta)$-differentially private if for all pairs of datasets $S, S' \in \mathcal{Z}^*$ that differs in exactly one data point, and every measurable $\mathcal{O} \subseteq \mathcal{R}$, we have*

$$\Pr\left(\mathcal{A}(S) \in \mathcal{O}\right) \leq e^\epsilon \cdot \Pr\left(\mathcal{A}(S') \in \mathcal{O}\right) + \delta,$$

*where the probability is over the random coins of $\mathcal{A}$. When $\delta = 0$, we say that $\mathcal{A}$ is* pure *$\epsilon$-differentially private.*

We study learning algorithms that take as input two datasets: a private dataset $S_{\mathsf{priv}}$ and a public dataset $S_{\mathsf{pub}}$, and output a hypothesis $h : \mathcal{X} \rightarrow \{0, 1\}$. The private set $S_{\mathsf{priv}} \in (\mathcal{X} \times \{0, 1\})^*$ is labeled. We distinguish between two settings of the learning problem depending on whether the public dataset is labeled or not. To avoid confusion, we denote an *unlabeled* public set as $T_{\mathsf{pub}} \in \mathcal{X}^*$, and use $S_{\mathsf{pub}}$ to denote a *labeled* public set. We formally define learners in these two settings.

**Definition 2.2** (($\alpha, \beta, \epsilon, \delta$)- Semi-Private Learner). *Let $\mathcal{H} \subset \{0, 1\}^{\mathcal{X}}$ be a hypothesis class. A randomized algorithm $\mathcal{A}$ is $(\alpha, \beta, \epsilon, \delta)$-SP (semi-private) learner for $\mathcal{H}$ with private sample size $n_{\mathsf{priv}}$ and public sample size $n_{\mathsf{pub}}$ if the following conditions hold:*

1. *For every distribution $\mathcal{D}$ over $\mathcal{Z} = \mathcal{X} \times \{0, 1\}$, given datasets $S_{\mathsf{priv}} \sim \mathcal{D}^{n_{\mathsf{priv}}}$ and $S_{\mathsf{pub}} \sim \mathcal{D}^{n_{\mathsf{pub}}}$ as inputs to $\mathcal{A}$, with probability at least $1 - \beta$ (over the choice of $S_{\mathsf{priv}}$, $S_{\mathsf{pub}}$, and the random coins of $\mathcal{A}$), $\mathcal{A}$ outputs a hypothesis $\mathcal{A}\,(S_{\mathsf{priv}}, S_{\mathsf{pub}}) = \hat{h} \in \{0, 1\}^{\mathcal{X}}$ satisfying*

$$\mathsf{err}\left(\hat{h};\, \mathcal{D}\right) \leq \inf_{h \in \mathcal{H}} \mathsf{err}\,(h;\, \mathcal{D}) + \alpha.$$

2. *For all $S \in \mathcal{Z}^{n_{\mathsf{pub}}}$, $\mathcal{A}\,(\cdot, S)$ is $(\epsilon, \delta)$-differentially private.*

*When the second condition is satisfied with $\delta = 0$ (i.e., pure differential privacy), we refer to $\mathcal{A}$ as $(\alpha, \beta, \epsilon)$-SP learner (i.e., pure semi-private learner).*

As a special case of the above definition, we say that an algorithm $\mathcal{A}$ is an $(\alpha, \beta, \epsilon, \delta)$-semi-privately learner for a class $\mathcal{H}$ under the realizability assumption if it satisfies the first condition in the definition only with respect to all distributions that are realizable by $\mathcal{H}$.

**Definition 2.3** (Semi-Privately Learnable Class). *We say that a class $\mathcal{H}$ is semi-privately learnable if there are functions $n_{\mathsf{priv}} : (0, 1)^2 \rightarrow \mathbb{N}$, $n_{\mathsf{pub}} : (0, 1)^2 \rightarrow \mathbb{N}$, where $n_{\mathsf{pub}}(\alpha, \cdot) = o(1/\alpha^2)$, and there is an algorithm $\mathcal{A}$ such that for every $\alpha, \beta \in (0, 1)$, when $\mathcal{A}$ is given private and public samples of sizes $n_{\mathsf{priv}} = n_{\mathsf{priv}}(\alpha, \beta)$, and $n_{\mathsf{pub}} = n_{\mathsf{pub}}(\alpha, \beta)$, it $(\alpha, \beta, 0.1, \mathsf{negl}\,(n_{\mathsf{priv}}))$-semi-privately learns $\mathcal{H}$.*

Note that in the definition above, the privacy parameters are set as follows: $\epsilon = 0.1$ and $\delta$ is negligible function in the private sample size (and $\delta = 0$ for a *pure* semi-private learner).

The restriction $n_{\mathsf{pub}} = o(1/\alpha^2)$ in the above definition is because taking $\Omega(\mathsf{VC}(\mathcal{H})/\alpha^2)$ public examples suffices for learning the class without any private examples.

**Definition 2.4** (($\alpha, \beta, \epsilon, \delta$)-Semi-Supervised Semi-Private Learner). *The definition is analogous to Definition 2.2 except that the public sample is* unlabeled. *An algorithm that satisfies this definition is referred to as $(\alpha, \beta, \epsilon, \delta)$-SS-SP (semi-supervised semi-private) learner.*

**Private learning without public data:** In the standard setting of $(\epsilon, \delta)$-differentially private learning, the learner has no access to public data. We note that this setting can be viewed as a special case of Definitions 2.2 and 2.4 by taking $n_{\mathsf{pub}} = 0$. In such case, we refer to the learner as $(\alpha, \beta, \epsilon, \delta)$-*private learner*. As before, when $\delta = 0$, we call the learner *pure private learner*. The notion of *privately learnable* class $\mathcal{H}$ is defined analogously to Definition 2.3 with $n_{\mathsf{pub}}(\alpha, \beta) = 0$ for all $\alpha, \beta$.

We will use the following lemma due to Beimel et al. [BNS15]:

**Lemma 2.5** (Special case of Theorem 4.16 in [BNS15]). *Any class $\mathcal{H}$ that is privately learnable under realizability assumption is also privately learnable (i.e., in the agnostic setting).*

The following fact follows from the private boosting technique due to [DRV10]:

**Lemma 2.6** (follows from Theorem 6.1 [DRV10] (the full version)). *For any class $\mathcal{H}$, under the realizability assumption, if there is a $(0.1, 0.1, 0.1)$-pure private learner for $\mathcal{H}$, then $\mathcal{H}$ is privately learnable by a pure private algorithm.*

We note that no analogous statement to the one in Lemma 2.6 is known for approximate private learners (see the full version [ABM19] for a discussion).

We will also use the following notion of coverings:

**Definition 2.7** ($\alpha$-cover for a hypothesis class). *A family of hypotheses $\widetilde{\mathcal{H}}$ is said to form an $\alpha$-cover for a hypothesis class $\mathcal{H} \subseteq \{0, 1\}^{\mathcal{X}}$ with respect to a distribution $\mathcal{D}_{\mathcal{X}}$ over $\mathcal{X}$ if for every $h \in \mathcal{H}$, there is $\tilde{h} \in \widetilde{\mathcal{H}}$ such that $\mathsf{dis}\left(h, \tilde{h}; \mathcal{D}_{\mathcal{X}}\right) \leq \alpha$.*

## 3   Upper Bound

In this section we show that every VC class $\mathcal{H}$ can be semi-privately learned in the agnostic case with only $\tilde{O}(\mathsf{VC}(\mathcal{H})/\alpha)$ public examples:

**Theorem 3.1** (Upper bound). *Let $\mathcal{H}$ be a hypothesis class and let $\mathsf{VC}(\mathcal{H}) = d$. For any $\alpha, \beta \in (0, 1)$, $\epsilon > 0$, $\mathcal{A}_{\mathsf{SSPP}}$ is an $(\alpha, \beta, \epsilon)$-semi-supervised semi-private agnostic learner for $\mathcal{H}$ with private and public sample complexities:*

$$n_{\mathsf{priv}} = O\left( \left(d \log(1/\alpha) + \log(1/\beta)\right) \max\left(\frac{1}{\alpha^2}, \frac{1}{\epsilon\, \alpha}\right)\right),$$

$$n_{\mathsf{pub}} = O\left(\frac{d \log(1/\alpha) + \log(1/\beta)}{\alpha}\right).$$

**Proof overview.** The upper bound is based on a reduction to the fact that any finite hypothesis class $\mathcal{H}'$ can be learned privately with sample complexity (roughly) $O(\log|\mathcal{H}'|)$ via the exponential mechanism [KLN+08]. In more detail, we use the (unlabeled) public data to construct a finite class $\mathcal{H}'$ that forms a "good enough *approximation*" of the (possibly infinite) original class $\mathcal{H}$ (See Algorithm 1). The relevant notion of approximation is captured by the definition of $\alpha$-cover (Definition 2.7). Indeed, it suffices to output an hypothesis $h' \in \mathcal{H}'$ that "$\alpha$-approximates" an optimal hypothesis $h^* \in \mathcal{H}$.

Thus, the crux of the proof boils down to the question: *How many samples from $\mathcal{D}_{\mathcal{X}}$ are needed in order to construct an $\alpha$-cover for $\mathcal{H}$?* It is not hard to see that (roughly) $O(\mathsf{VC}(\mathcal{H})/\alpha^2)$ examples suffice: indeed, these many examples suffice to approximate the distances $\mathsf{dis}(h', h''; \mathcal{D}_{\mathcal{X}})$ for every $h', h'' \in \mathcal{H}$, which suffices to construct the $\alpha$-cover. We show how to reduce the number of examples to only (roughly) $O(\mathsf{VC}(\mathcal{H})/\alpha)$ examples (Lemma 3.3), which, by our lower bound, is nearly optimal.

---

**Algorithm 1** $\mathcal{A}_{\mathsf{SSPP}}$: Semi-Supervised Semi-Private Agnostic Learner

---

**Input:** Private labeled dataset: $S_{\mathsf{priv}} = \{(x_1, y_1), \ldots, (x_{n_{\mathsf{priv}}}, y_{n_{\mathsf{priv}}})\} \in \mathcal{Z}^{n_{\mathsf{priv}}}$, a public unlabeled dataset: $T_{\mathsf{pub}} = (\tilde{x}_1, \cdots, \tilde{x}_{n_{\mathsf{pub}}}) \in \mathcal{X}^{n_{\mathsf{pub}}}$, a hypothesis class $\mathcal{H} \subset \{0, 1\}^{\mathcal{X}}$, and a privacy parameter $\epsilon > 0$.
1: Let $\widetilde{T} = \{\hat{x}_1, \ldots, \hat{x}_{\hat{m}}\}$ be the set of points $x \in \mathcal{X}$ appearing at least once in $T_{\mathsf{pub}}$.
2: Let $\Pi_{\mathcal{H}}(\widetilde{T}) = \{(h(\hat{x}_1), \ldots, h(\hat{x}_{\hat{m}})) : h \in \mathcal{H}\}$.
3: Initialize $\widetilde{\mathcal{H}}_{T_{\mathsf{pub}}} = \emptyset$.
4: **for** each $\mathbf{c} = (c_1, \ldots, c_{\hat{m}}) \in \Pi_{\mathcal{H}}(\widetilde{T})$: **do**
5:    Add to $\widetilde{\mathcal{H}}_{T_{\mathsf{pub}}}$ arbitrary $h \in \mathcal{H}$ that satisfies $h(\hat{x}_j) = c_j$ for every $j = 1, \ldots, \hat{m}$.
6: Use the exponential mechanism with inputs $S_{\mathsf{priv}}, \widetilde{\mathcal{H}}_{T_{\mathsf{pub}}}, \epsilon$ and score function $q(S_{\mathsf{priv}}, h) \triangleq -\widehat{\mathsf{err}}(h; S_{\mathsf{priv}})$ to select $h_{\mathsf{priv}} \in \widetilde{\mathcal{H}}_{T_{\mathsf{pub}}}$.
7: **return** $h_{\mathsf{priv}}$.

---

The proof of Theorem 3.1 relies on the following lemmas. (The full proof of Theorem 3.1 can be found in the full version [ABM19]).

**Lemma 3.2.** *For all $T_{\mathsf{pub}} \in \mathcal{X}^{n_{\mathsf{pub}}}$, $\mathcal{A}_{\mathsf{SSPP}}(\cdot,\ T_{\mathsf{pub}})$ is $\epsilon$-differentially private.*

The proof is straightforward and is deferred to the full version [ABM19].

**Lemma 3.3** ($\alpha$-cover for $\mathcal{H}$). *Let $T_{\mathsf{pub}} \sim \mathcal{D}_{\mathcal{X}}^{n_{\mathsf{pub}}}$, where $n_{\mathsf{pub}} = O\left(\frac{d\ \log(1/\alpha) + \log(1/\beta)}{\alpha}\right)$. Then, with probability at least $1 - \beta$, the family $\widetilde{\mathcal{H}}_{T_{\mathsf{pub}}}$ constructed in Step 5 of Algorithm 1 is an $\alpha$-cover for $\mathcal{H}$ w.r.t. $\mathcal{D}_{\mathcal{X}}$.*

We now prove Lemma 3.3. (We include a a more detailed version in the full version [ABM19]). We need to show that with high probability, for every $h \in \mathcal{H}$ there exists $\tilde{h} \in \widetilde{\mathcal{H}}_{T_{\mathsf{pub}}}$ such that $\mathsf{dis}(h, \tilde{h}; \mathcal{D}_{\mathcal{X}}) \leq \alpha$. Let $\widetilde{T} = \{\hat{x}_1, \ldots, \hat{x}_{\hat{m}}\}$ be as defined in $\mathcal{A}_{\mathsf{SSPP}}$ (Algorithm 1), and define $h(\widetilde{T}) = (h(\hat{x}_1), \ldots, h(\hat{x}_{\hat{m}}))$. By construction, there must exist $\tilde{h} \in \widetilde{\mathcal{H}}_{T_{\mathsf{pub}}}$ such that $\forall j \in [\hat{m}]$ $\tilde{h}(\hat{x}_j) = h(\hat{x}_j)$; that is, $\widehat{\mathsf{dis}}\left(\tilde{h}, h;\ T_{\mathsf{pub}}\right) = 0$. For $T_{\mathsf{pub}} \sim \mathcal{D}_{\mathcal{X}}^{n_{\mathsf{pub}}}$, define the event

$$\mathsf{Bad} = \left\{\exists h_1, h_2 \in \mathcal{H} :\ \mathsf{dis}\,(h_1, h_2; \mathcal{D}_{\mathcal{X}}) > \alpha\ \text{ and }\ \widehat{\mathsf{dis}}\,(h_1, h_2;\ T_{\mathsf{pub}}) = 0\right\}$$

We will show that

$$\mathbb{P}_{T_{\mathsf{pub}} \sim \mathcal{D}_{\mathcal{X}}^{n_{\mathsf{pub}}}} [\mathsf{Bad}] \leq 2\left(\frac{2e\,n_{\mathsf{pub}}}{d}\right)^{2d} e^{-\alpha\,n_{\mathsf{pub}}/4}. \tag{1}$$

Before we do so, we first show that (1) suffices to prove the lemma. Indeed, if $\mathsf{dis}\left(\tilde{h}, h;\ \mathcal{D}_{\mathcal{X}}\right) > \alpha$ for some $h \in \mathcal{H}$ then the event Bad occurs. Hence,

$$\mathbb{P}_{T_{\mathsf{pub}} \sim \mathcal{D}_{\mathcal{X}}^{n_{\mathsf{pub}}}} \left[\widetilde{\mathcal{H}}_{T_{\mathsf{pub}}} \text{ is not an } \alpha\text{-cover}\right] \leq 2\left(\frac{2e\,n_{\mathsf{pub}}}{d}\right)^{2d} e^{-\alpha\,n_{\mathsf{pub}}/4}.$$

Now, via standard manipulation, this bound is at most $\beta$ when $n_{\mathsf{pub}} = O\left(\frac{d\ \log(1/\alpha) + \log(1/\beta)}{\alpha}\right)$, which yields the desired bound and finishes the proof.

Now, it is left to prove (1). To do so, we use a standard VC-based uniform convergence bound (a.k.a $\alpha$-net bound) on the class $\mathcal{H}_{\Delta} \triangleq \{h_1 \Delta h_2 : h_1, h_2 \in \mathcal{H}\}$ where $h_1 \Delta h_2 : \mathcal{X} \to \{0, 1\}$ is defined as

$$h_1 \Delta h_2(x) \triangleq \mathbf{1}\,(h_1(x) \neq h_2(x)) \quad \forall x \in \mathcal{X}$$

Let $\mathcal{G}_{\mathcal{H}_{\Delta}}$ denote the growth function of $\mathcal{H}_{\Delta}$; i.e., for any $m$, $\mathcal{G}_{\mathcal{H}_{\Delta}}(m) \triangleq \max_{V:|V|=m} |\Pi_{\mathcal{H}_{\Delta}}(V)|$, where $\Pi_{\mathcal{H}_{\Delta}}(V)$ is the set of all possible dichotomies that can be generated by $\mathcal{H}_{\Delta}$ on a set $V$ of size $m$. Note that $\mathcal{G}_{\mathcal{H}_{\Delta}}(m) \leq \left(\frac{e\,m}{d}\right)^{2d}$. This follows from the fact that for any set $V$ of size $m$, we have $|\Pi_{\mathcal{H}_{\Delta}}(V)| \leq |\Pi_{\mathcal{H}}(V)|^2$ since every dichotomy in $\Pi_{\mathcal{H}_{\Delta}}$ is determined by a pair of dichotomies in $\Pi_{\mathcal{H}}(V)$. Hence, $\mathcal{G}_{\mathcal{H}_{\Delta}}(m) \leq (\mathcal{G}_{\mathcal{H}}(m))^2 \leq \left(\frac{e\,m}{d}\right)^{2d}$, where the last inequality follows from Sauer's Lemma [Sau72]. Now, by invoking a uniform convergence argument, we have

$$\mathbb{P}_{T_{\mathsf{pub}} \sim \mathcal{D}_{\mathcal{X}}^{n_{\mathsf{pub}}}} [\mathsf{Bad}] = \mathbb{P}_{T_{\mathsf{pub}} \sim \mathcal{D}_{\mathcal{X}}^{n_{\mathsf{pub}}}} \left[\exists h \in \mathcal{H}_{\Delta} :\ \mathsf{dis}\,(h, h_{\mathbf{0}};\ \mathcal{D}_{\mathcal{X}}) > \alpha\ \text{ and }\ \widehat{\mathsf{dis}}\,(h, h_{\mathbf{0}};\ T_{\mathsf{pub}}) = 0\right]$$

$$\leq 2\mathcal{G}_{\mathcal{H}_{\Delta}}(2\,n_{\mathsf{pub}})\,e^{-\alpha\,n_{\mathsf{pub}}/4} \leq 2\left(\frac{2e\,n_{\mathsf{pub}}}{d}\right)^{2d} e^{-\alpha\,n_{\mathsf{pub}}/4}.$$

The first bound in the second line follows from the so-called double-sample argument used in virtually all VC-based uniform convergence bounds (e.g., [SSBD14]). This completes the proof of Lemma 3.3.

## 4 Lower Bound

In this section we establish that our upper bound on the public sample complexity is nearly tight.

**Theorem 4.1** (Lower bound for classes of infinite Littlestone dimension). *Let $\mathcal{H}$ be any class with an infinite Littlestone dimension (e.g., the class of thresholds over $\mathbb{R}$). Then, any semi-private learner for $\mathcal{H}$ must have public sample of size $n_{\mathsf{pub}} = \Omega(1/\alpha)$, where $\alpha$ is the excess error.*

In the case of pure differentially privacy we get a stronger statement which manifests a dichotomy that applies for every class:

**Theorem 4.2** (Pure private vs. pure semi-private learners). *Every class $\mathcal{H}$ must satisfy exactly one of the following:*

1. *$\mathcal{H}$ is learnable by a pure private learner.*

2. *Any pure semi-private learner for $\mathcal{H}$ must have $n_{\mathsf{pub}} = \Omega(1/\alpha)$, where $\alpha$ is the excess error*

**Proof overview.** The crux of the argument is a *public-data-reduction lemma* (Lemma 4.4), which shows how one can reduce the number of public examples at the price of a proportional increase in the excess error. This lemma implies, for example, that if $\mathcal{H}$ can be learned up to an excess error of $\alpha$ with less than $\frac{1}{1000\alpha}$ public examples then it can also be privately learned without any public examples and excess error $< \frac{1}{10}$. Stating contra-positively, if $\mathcal{H}$ can not be privately learned with excess error $< \frac{1}{10}$ then it can not be semi-privately learned with excess error of $\alpha$ with less than $\frac{1}{1000\alpha}$ public examples. This yields a lower bound of $\Omega(1/\alpha)$ on the public sample complexity for every class $\mathcal{H}$ which is not privately learnable with constant excess error

One example for such a class is any class with infinite Littlestone dimension (e.g., the class of 1-dimensional thresholds over an infinite domain). This follows from the result in [ALMM19]:

**Theorem 4.3** (Restatement of Corollary 2 in [ALMM19]). *Let $\mathcal{H}$ be any class of infinite Littlestone dimension (e.g., the class of thresholds over an infinite domain $\mathcal{X} \subseteq \mathbb{R}$). For any $n \in \mathbb{N}$, given a private sample of size $n$, there is no $\left( \frac{1}{16}, \frac{1}{16}, 0.1, \frac{1}{100\,n^2\log(n)} \right)$-private learner for $\mathcal{H}$ (even in the realizable case).*

The aforementioned reduction we use for the lower bound holds even when the public sample is *labeled*, and it holds for both *pure* and *approximate* private/semi-private learners.

We now state and prove the reduction lemma outlined above.

**Lemma 4.4** (Public data reduction lemma). *Let $0 < \alpha \le 1/100$, $\epsilon > 0$, $\delta \ge 0$. Suppose there is an $(\alpha, \frac{1}{18}, \epsilon, \delta)$-agnostic semi-private learner for an hypothesis class $\mathcal{H}$ with private sample size $n_{\mathsf{priv}}$ and public sample size $n_{\mathsf{pub}}$. Then, there is a $\left(100\,n_{\mathsf{pub}}\,\alpha, \frac{1}{16}, \epsilon, \delta\right)$-private learner that learns any distribution realizable by $\mathcal{H}$ with input sample size $\lceil \frac{n_{\mathsf{priv}}}{10\,n_{\mathsf{pub}}} \rceil$.*

*Proof.* Let $\mathcal{A}$ denote the assumed agnostic-case semi-private learner for $\mathcal{H}$ with input private sample of size $n_{\mathsf{priv}}$ and input public sample of size $n_{\mathsf{pub}}$. Using $\mathcal{A}$, we construct a realizable-case private learner for $\mathcal{H}$, which we denote by $\mathcal{B}$. The description of $\mathcal{B}$ appears in Algorithm 2.

The following two claims about $\mathcal{B}$ suffice to prove the lemma.

---
**Algorithm 2** Description of the private learner $\mathcal{B}$:

---
**Input:** Private sample $\tilde{S} = (\tilde{z}_1, \ldots, \tilde{z}_{\tilde{n}})$ of size $\tilde{n} = \lceil n_{\mathsf{priv}}/(10 \cdot n_{\mathsf{pub}}) \rceil$.
 1: Pick a fixed (dummy) distribution $\mathcal{D}_0$ over $\mathcal{Z} = \mathcal{X} \times \{0, 1\}$ where the label $y \in \{0, 1\}$ is drawn uniformly at random from $\{0, 1\}$ independently from $x \in \mathcal{X}$.
 2: Set $p = 1/(100 \cdot n_{\mathsf{pub}})$.
 3: Using $\tilde{S}$ and $\mathcal{D}_0$, construct samples $S_{\mathsf{priv}}$, $S_{\mathsf{pub}}$ using procedures $\mathsf{PrivSamp}(\tilde{S}, \mathcal{D}_0, p, n_{\mathsf{priv}})$ and $\mathsf{PubSamp}(\mathcal{D}_0, n_{\mathsf{pub}})$ given by Algorithms 3 and 4 below.
 4: Return $\tilde{h} = \mathcal{A}(S_{\mathsf{priv}}, S_{\mathsf{pub}})$.

---

**Claim 4.5** (Privacy guarantee of $\mathcal{B}$). *$\mathcal{B}$ is $(\epsilon, \delta)$-differentially private*

The above claim easily follows since $\mathcal{A}$ is a semi-private learner, $S_{\mathsf{pub}}$ does not contain any points from $\tilde{S}$, and each point in $\tilde{S}$ appears at most once in $S_{\mathsf{priv}}$.

**Claim 4.6** (Accuracy guarantee of $\mathcal{B}$). *Let $\mathcal{D}$ be any distribution over $\mathcal{Z}$ that is realizable by $\mathcal{H}$. Suppose $\tilde{S} \sim \mathcal{D}^{\tilde{n}}$. Then, except with probability at most $1/16$ (over the choice of $\tilde{S}$ and internal randomness in $\mathcal{B}$), the output hypothesis $\tilde{h}$ satisfies: $\mathsf{err}(\tilde{h}; \mathcal{D}) \le 100\,n_{\mathsf{pub}}\,\alpha$.*

---
**Algorithm 3** Private Sample Generator PrivSamp:
---
**Input:** Sample $\tilde{S} = (\tilde{z}_1, \ldots, \tilde{z}_{\tilde{n}})$, Distribution $\mathcal{D}_0$, parameter $p$, sample size $n_{\mathsf{priv}}$.
1: $i := 1$
2: **while** $\tilde{S} \neq \emptyset$ **and** $i \leq n_{\mathsf{priv}}$: **do**
3:     Sample $b_i \sim \mathsf{Ber}(p)$ (independently for each $i$), where $\mathsf{Ber}(p)$ is Bernoulli distribution with mean $p$.
4:     **if** $b_i = 1$: **then**
5:         Set $z_i^{\mathsf{prv}}$ to be the next element in $\tilde{S}$, i.e., $z_i^{\mathsf{prv}} = \tilde{z}_{j_i}$, where $j_i = \sum_{k=1}^{i} b_k$.
6:         Remove this element from $\tilde{S}$: $\tilde{S} \leftarrow \tilde{S} \setminus \tilde{z}_{j_i}$.
7:     **else**
8:         Set $z_i^{\mathsf{prv}} = z_i^0$, where $z_i^0$ is a fresh independent example from the "dummy" distribution $\mathcal{D}_0$.
9:     $i \leftarrow i + 1$
10: **return** $S_{\mathsf{priv}} = (z_1^{\mathsf{prv}}, \ldots, z_{n_{\mathsf{priv}}}^{\mathsf{prv}})$.
---

---
**Algorithm 4** Public Sample Generator PubSamp:
---
**Input:** Distribution $\mathcal{D}_0$, sample size $n_{\mathsf{pub}}$.
1: **for** $i = 1, \ldots, n_{\mathsf{pub}}$ : **do**
2:     Set $z_i^{\mathsf{pub}} = z_i^0$ where $z_i^0$ is a fresh independent example from $\mathcal{D}_0$.
3: **return** $S_{\mathsf{pub}} = (z_1^{\mathsf{pub}}, \ldots, z_{n_{\mathsf{pub}}}^{\mathsf{pub}})$
---

Let $\mathcal{D}_{(p)}$ denote the mixture distribution $p \cdot \mathcal{D} + (1-p) \cdot \mathcal{D}_0$ (recall the definition of $p$ from Algorithm 2). To prove Claim 4.6, we first show that both $S_{\mathsf{priv}}$ and $S_{\mathsf{pub}}$ can be "viewed" as being sampled from $\mathcal{D}_{(p)}$. The claim will then follow since $\mathcal{A}$ learns $\mathcal{H}$ with respect to $\mathcal{D}_{(p)}$.

First, note that since $\tilde{n} = 10 \cdot p \cdot n_{\mathsf{priv}}$, then by Chernoff's bound, except with probability $< 0.01$, Algorithm 3 exits the **WHILE** loop with $i = n_{\mathsf{priv}}$. Thus, except with probability $< 0.01$, we have

$$|S_{\mathsf{priv}}| = n_{\mathsf{priv}}, \text{ hence, } S_{\mathsf{priv}} \sim \mathcal{D}_{(p)}^{n_{\mathsf{priv}}}. \tag{2}$$

As for $S_{\mathsf{pub}}$, note that $S_{\mathsf{pub}} = (z_1^0, \ldots, z_{n_{\mathsf{pub}}}^0) \sim \mathcal{D}_0^{n_{\mathsf{pub}}}$. We will show that $\mathcal{D}_0^{n_{\mathsf{pub}}}$ is close in total variation to $\mathcal{D}_{(p)}^{n_{\mathsf{pub}}}$. Let $\widehat{S}_{\mathsf{pub}} = (\hat{z}_1, \ldots, \hat{z}_{n_{\mathsf{pub}}})$ be i.i.d. sequence generated as follows: for each $i \in [n_{\mathsf{pub}}]$, $\hat{z}_i = b_i \, v_i + (1 - b_i) \, z_i^0$, where $(b_1, \ldots, b_{n_{\mathsf{pub}}}) \sim (\mathsf{Ber}(p))^{n_{\mathsf{pub}}}$, and $(v_1, \ldots, v_n) \sim \mathcal{D}^{n_{\mathsf{pub}}}$. It is clear that $\widehat{S}_{\mathsf{pub}} \sim \mathcal{D}_{(p)}^{n_{\mathsf{pub}}}$. Moreover, observe that

$$\mathbb{P}\left[\widehat{S}_{\mathsf{pub}} = S_{\mathsf{pub}}\right] \geq \mathbb{P}\left[b_i = 0 \;\; \forall \, i \in [n_{\mathsf{pub}}]\right] = \left(1 - \frac{1}{100 \, n_{\mathsf{pub}}}\right)^{n_{\mathsf{pub}}} \geq 0.99$$

Note that $\mathbb{P}\left[\widehat{S}_{\mathsf{pub}} \neq S_{\mathsf{pub}}\right]$ is the probability measure attributed to the first component of the mixture distribution $\mathcal{D}_{(p)}$ of $\hat{S}_{pub}$ (i.e., the component from $\mathcal{D}$). Hence, it follows that the total variation between the distribution of $\hat{S}_{pub}$ (induced by the mixture $\mathcal{D}_{(p)}$) and the distribution of $S_{pub}$ (induced by $\mathcal{D}_0$) is at most $0.01$. In particular, the probability of any event w.r.t. the distribution of $\widehat{S}_{\mathsf{pub}}$ is at most $0.01$ far from the probability of same event w.r.t. the distribution of $S_{\mathsf{pub}}$. Hence,

$$\mathbb{P}_{S_{\mathsf{priv}}, S_{\mathsf{pub}}, \mathcal{A}}\left[\mathsf{err}\left(\mathcal{A}(S_{\mathsf{priv}}, S_{\mathsf{pub}}); \; \mathcal{D}_{(p)}\right) - \min_{h \in \mathcal{H}} \mathsf{err}(h; \mathcal{D}_{(p)}) > \alpha\right]$$
$$- \mathbb{P}_{S_{\mathsf{priv}}, \widehat{S}_{\mathsf{pub}}, \mathcal{A}}\left[\mathsf{err}\left(\mathcal{A}(S_{\mathsf{priv}}, S_{\mathsf{pub}}); \; \mathcal{D}_{(p)}\right) - \min_{h \in \mathcal{H}} \mathsf{err}(h; \mathcal{D}_{(p)}) > \alpha\right] \leq 0.01 \tag{3}$$

Now, from (2) and the premise that $\mathcal{A}$ is agnostic semi-private learner, we have

$$\mathbb{P}_{S_{\mathsf{priv}}, \widehat{S}_{\mathsf{pub}}, \mathcal{A}}\left[\mathsf{err}\left(\mathcal{A}(S_{\mathsf{priv}}, S_{\mathsf{pub}}); \; \mathcal{D}_{(p)}\right) - \min_{h \in \mathcal{H}} \mathsf{err}(h; \mathcal{D}_{(p)}) > \alpha\right] \leq \frac{1}{17}$$

Hence, using (3), we conclude that except with probability $< 1/16$,

$$\mathsf{err}\left(\mathcal{A}(S_{\mathsf{priv}}, S_{\mathsf{pub}}); \; \mathcal{D}_{(p)}\right) - \min_{h \in \mathcal{H}} \mathsf{err}(h; \mathcal{D}_{(p)}) \leq \alpha. \tag{4}$$

Note that for any $h$, $\mathsf{err}(h;\ \mathcal{D}_{(p)}) = p\cdot\mathsf{err}(h;\ \mathcal{D}) + (1-p)\cdot\mathsf{err}(h;\mathcal{D}_0) = p\cdot\mathsf{err}(h;\ \mathcal{D}) + \frac{1}{2}(1-p)$, where the last equality follows from the fact that the labels generated by $\mathcal{D}_0$ are completely noisy (uniformly random labels). Hence, we have $\arg\min_{h\in\mathcal{H}}\mathsf{err}(h;\mathcal{D}_{(p)}) = \arg\min_{h\in\mathcal{H}}\mathsf{err}(h;\mathcal{D})$. That is, the optimal hypothesis with respect to the realizable distribution $\mathcal{D}$ is also optimal with respect to the mixture distribution $\mathcal{D}_{(p)}$. Let $h^* \in \mathcal{H}$ denote such hypothesis. Note that $\mathsf{err}(h^*;\mathcal{D}) = 0$ and $\mathsf{err}(h^*;\mathcal{D}_{(p)}) = \frac{1}{2}(1-p)$. These observations together with (4) imply that except with probability $< 1/16$, we have

$$\alpha \geq p\cdot\mathsf{err}\left(\mathcal{A}(S_{\mathsf{priv}}, S_{\mathsf{pub}});\ \mathcal{D}\right)$$

Hence, $\mathsf{err}\left(\mathcal{B}(\tilde{S});\ \mathcal{D}\right) = \mathsf{err}\left(\mathcal{A}(S_{\mathsf{priv}}, S_{\mathsf{pub}});\ \mathcal{D}\right) \leq 100\cdot n_{\mathsf{pub}}\cdot\alpha$. This completes the proof.

$\square$

With Lemma 4.4, we are now ready to prove the main results for this section:

**Proof of Theorem 4.1**

*Proof.* Suppose $\mathcal{A}$ is a semi-private learner for $\mathcal{H}$ with sample complexities $n_{\mathsf{priv}}, n_{\mathsf{pub}}$. In particular, given $n_{\mathsf{priv}}(\alpha, \frac{1}{18}), n_{\mathsf{pub}}(\alpha, \frac{1}{18})$ private and public examples, $\mathcal{A}$ is $(\alpha, \frac{1}{18}, 0.1, \frac{1}{100\, n_{\mathsf{priv}}^2\, \log(n_{\mathsf{priv}})})$-semi-private learner for $\mathcal{H}$. Hence, by Lemma 4.4, there is $(100 n_{\mathsf{pub}}\alpha, \frac{1}{16}, 0.1, \frac{1}{100\, n_{\mathsf{priv}}^2\, \log(n_{\mathsf{priv}})})$-private learner for $\mathcal{H}$. Thus, Theorem 4.3 implies that $100 n_{\mathsf{pub}}\alpha > \frac{1}{16}$ and hence that $n_{\mathsf{pub}} > \frac{1}{1600\,\alpha}$ as required. $\square$

**Proof of Theorem 4.2**

*Proof.* First, if $\mathcal{H}$ is learnable by a pure private learner, then trivially the second condition cannot hold since $\mathcal{H}$ can be learned without any public examples. Now, suppose that the first item does *not* hold. Note that by Lemma 2.5, this implies that there is *no* pure private learner for $\mathcal{H}$ with respect to realizable distributions. By Lemma 2.6, this in turn implies that there is *no* $\left(\frac{1}{16}, \frac{1}{16}, 0.1\right)$-pure private learner for $\mathcal{H}$ with respect to realizable distributions. Now, suppose $\mathcal{A}$ is a pure semi-private learner $\mathcal{A}$ for $\mathcal{H}$. Then, this implies that for any $\alpha > 0$, $\mathcal{A}$ is an $\left(\alpha, \frac{1}{18}, 0.1\right)$-pure semi-private learner for $\mathcal{H}$ with sample complexities $n_{\mathsf{priv}}(\alpha, \frac{1}{18}), n_{\mathsf{pub}}(\alpha, \frac{1}{18})$. Hence, by Lemma 4.4, there is a $\left(100\, n_{\mathsf{pub}}\, \alpha, \frac{1}{16}, 0.1\right)$-pure private learner for $\mathcal{H}$ w.r.t. realizable distributions. This together with the earlier conclusion implies that $100\, n_{\mathsf{pub}}\, \alpha > \frac{1}{16}$, and therefore that $n_{\mathsf{pub}} > \frac{1}{1600\,\alpha}$, which shows that the condition in the second item holds. $\square$

### Acknowledgements

N. Alon's research is supported in part by NSF grant DMS-1855464, BSF grant 2018267, and the Simons Foundation. R. Bassily's research is supported by NSF Awards AF-1908281, SHF-1907715, Google Faculty Research Award, and OSU faculty start-up support.

## Footnotes

[2]The Littlestone dimension is a combinatorial parameter that arises in online learning [Lit87, BPS09].

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
