[Supplementary Material · Limits_of_private_learning_with_public_data_supp.pdf]

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

**Related Work**

The most related work to ours is the work of Beimel et al. [BNS13], which focuses on the realizable case of semi-private learning and give an upper bound on the sample complexity. The algorithm we use in our upper bound is essentially the same as theirs. However, our analysis for the sample complexity differs from theirs: our argument relies on the notion of $\alpha$-coverings, which provides a direct argument that extends to the agnostic case.

There are also several other works that considered similar problems. A similar notion known as "label-private learning" was considered in [CH11] (see also references therein) and in [BNS13]. In this notion, only the labels in the training set are considered private. This notion is weaker than semi-private learning. In particular, any semi-private learner can be easily transformed into a label-private learner. Another line of work consider the problem of private knowledge transfer [HCB16], [PAE+17], [PSM+18], and [BTT18]. In this problem, first a DP classification algorithm with input private sample is used to provide labels for an unlabeled public dataset. Then, the resulting dataset is used to train a non-private learner. The work of [BTT18] gives upper bounds on private and public sample complexities in the setting when the DP algorithm is required to label the public data in an online fashion. Their bounds are thus not comparable to ours.

# 2 Preliminaries

## Notation

For $n \in \mathbb{N}$, we use $[n]$ to denote the set $\{1, \ldots, n\}$. We use standard asymptotic notation $O, \Omega, o, \omega$. A function $f : \mathbb{N} \to [0, 1]$ is said to be negligible if $f(n) = o(n^{-d})$ for every $d \in \mathbb{N}$. The statement "$f$ is negligible" is denoted by $f = \mathsf{negl}(n)$.

We use standard notation from the supervised learning literature (see, e.g. [SSBD14]). Let $\mathcal{X}$ denote an arbitrary domain, let $\mathcal{Z} = \mathcal{X} \times \{0, 1\}$ denote the examples domain, and let $\mathcal{Z}^* = \cup_{n=1}^{\infty} \mathcal{Z}^n$. A function $h : \mathcal{X} \to \{0, 1\}$ is called a concept/hypothesis, a set of hypotheses $\mathcal{H} \subseteq \{0, 1\}^{\mathcal{X}}$ is called a concept/hypothesis class. The VC dimension of $\mathcal{H}$ is denoted by $\mathsf{VC}(\mathcal{H})$. We use $\mathcal{D}$ to denote a distribution over $\mathcal{Z}$, and $\mathcal{D}_{\mathcal{X}}$ to denote the marginal distribution over $\mathcal{X}$. We use $S \sim \mathcal{D}^n$ to denote a sample/dataset $S = \{(x_1, y_1), \ldots, (x_n, y_n)\}$ of $n$ i.i.d. draws from $\mathcal{D}$.

**Expected error:** The expected/population error of a hypothesis $h : \mathcal{X} \to \{0, 1\}$ with respect to a distribution $\mathcal{D}$ over $\mathcal{Z}$ is defined by $\mathsf{err}(h; \mathcal{D}) \triangleq \mathop{\mathbb{E}}_{(x,y) \sim \mathcal{D}} [\mathbf{1}\,(h(x) \neq y)]$.

A distribution $\mathcal{D}$ is called *realizable by* $\mathcal{H}$ if there exists $h^* \in \mathcal{H}$ such that $\mathsf{err}(h^*; \mathcal{D}) = 0$. In this case, the data distribution $\mathcal{D}$ is described by a distribution $\mathcal{D}_{\mathcal{X}}$ over $\mathcal{X}$ and a hypothesis $h^* \in \mathcal{H}$. For realizable distributions, the expected error of a hypothesis $h$ will be denoted by $\mathsf{err}\,(h;\,(\mathcal{D}_{\mathcal{X}}, h^*)) \triangleq \mathop{\mathbb{E}}_{x \sim \mathcal{D}_{\mathcal{X}}} [\mathbf{1}\,(h(x) \neq h^*(x))]$.

**Empirical error:** The empirical error of an hypothesis $h : \mathcal{X} \to \{0, 1\}$ with respect to a labeled dataset $S = \{(x_1, y_1), \ldots, (x_n, y_n)\}$ will be denoted by $\widehat{\mathsf{err}}\,(h; S) \triangleq \frac{1}{n} \sum_{i=1}^{n} \mathbf{1}\,(h(x_i) \neq y_i)$.

**Expected disagreement:** The expected disagreement between a pair of hypotheses $h_1$ and $h_2$ with respect to a distribution $\mathcal{D}_{\mathcal{X}}$ over $\mathcal{X}$ is defined as $\mathsf{dis}\,(h_1, h_2;\, \mathcal{D}_{\mathcal{X}}) \triangleq \mathop{\mathbb{E}}_{x \sim \mathcal{D}_{\mathcal{X}}} [\mathbf{1}\,(h_1(x) \neq h_2(x))]$.

**Empirical disagreement:** The empirical disagreement between a pair of hypotheses $h_1$ and $h_2$ w.r.t. an unlabeled dataset $T = \{x_1, \ldots, x_n\}$ is defined as $\widehat{\mathsf{dis}}\,(h_1, h_2;\, T) = \frac{1}{n} \sum_{i=1}^{n} \mathbf{1}\,(h_1(x_i) \neq h_2(x_i))$.

## Definitions

**Definition 2.1** (Differential Privacy [DMNS06, DKM+06]). *Let $\epsilon, \delta > 0$. A (randomized) algorithm $\mathcal{A}$ with input domain $\mathcal{Z}^*$ and output range $\mathcal{R}$ is called $(\epsilon, \delta)$-differentially private if for all pairs of datasets $S, S' \in \mathcal{Z}^*$ that differs in exactly one data point, and every measurable $\mathcal{O} \subseteq \mathcal{R}$, we have*

$$\Pr\,(\mathcal{A}(S) \in \mathcal{O}) \leq e^{\epsilon} \cdot \Pr\,(\mathcal{A}(S') \in \mathcal{O}) + \delta,$$

*where the probability is over the random coins of $\mathcal{A}$. When $\delta = 0$, we say that $\mathcal{A}$ is* pure $\epsilon$-*differentially private.*

We study learning algorithms that take as input two datasets: a private dataset $S_{\mathsf{priv}}$ and a public dataset $S_{\mathsf{pub}}$, and output a hypothesis $h : \mathcal{X} \to \{0, 1\}$. The public set entails no privacy constraint, whereas the algorithm is required to satisfy differential privacy with respect to $S_{\mathsf{priv}}$. The private set $S_{\mathsf{priv}} \in (\mathcal{X} \times \{0, 1\})^*$ is labeled. We distinguish between two settings of the learning problem depending on whether the public dataset is labeled or not. To avoid confusion, we will usually denote an *unlabeled* public set as $T_{\mathsf{pub}} \in \mathcal{X}^*$, and use $S_{\mathsf{pub}}$ to denote a *labeled* public set. We formally define learners in these two settings.

**Definition 2.2** (($\alpha, \beta, \epsilon, \delta$)- Semi-Private Learner)**.** *Let $\mathcal{H} \subset \{0, 1\}^{\mathcal{X}}$ be a hypothesis class. A randomized algorithm $\mathcal{A}$ is ($\alpha, \beta, \epsilon, \delta$)-SP (semi-private) learner for $\mathcal{H}$ with private sample size $n_{\mathsf{priv}}$ and public sample size $n_{\mathsf{pub}}$ if the following conditions hold:*

1. *For every distribution $\mathcal{D}$ over $\mathcal{Z} = \mathcal{X} \times \{0, 1\}$, given datasets $S_{\mathsf{priv}} \sim \mathcal{D}^{n_{\mathsf{priv}}}$ and $S_{\mathsf{pub}} \sim \mathcal{D}^{n_{\mathsf{pub}}}$ as inputs to $\mathcal{A}$, with probability at least $1 - \beta$ (over the choice of $S_{\mathsf{priv}}$, $S_{\mathsf{pub}}$, and the random coins of $\mathcal{A}$), $\mathcal{A}$ outputs a hypothesis $\mathcal{A}(S_{\mathsf{priv}}, S_{\mathsf{pub}}) = \hat{h} \in \{0, 1\}^{\mathcal{X}}$ satisfying*

$$\mathsf{err}\left(\hat{h};\ \mathcal{D}\right) \le \inf_{h \in \mathcal{H}} \mathsf{err}\left(h;\ \mathcal{D}\right) + \alpha.$$

2. *For all $S \in \mathcal{Z}^{n_{\mathsf{pub}}}$, $\mathcal{A}(\cdot, S)$ is ($\epsilon, \delta$)-differentially private.*

*When the second condition is satisfied with $\delta = 0$ (i.e., pure differential privacy), we refer to $\mathcal{A}$ as ($\alpha, \beta, \epsilon$)-SP learner (i.e., pure semi-private learner).*

As a special case of the above definition, we say that an algorithm $\mathcal{A}$ is an ($\alpha, \beta, \epsilon, \delta$)-semi-privately learner for a class $\mathcal{H}$ under the realizability assumption if it satisfies the first condition in the definition only with respect to all distributions that are realizable by $\mathcal{H}$.

**Definition 2.3** (Semi-Privately Learnable Class)**.** *We say that a class $\mathcal{H}$ is semi-privately learnable if there are functions $n_{\mathsf{priv}} : (0, 1)^2 \to \mathbb{N}$, $n_{\mathsf{pub}} : (0, 1)^2 \to \mathbb{N}$, where $n_{\mathsf{pub}}(\alpha, \cdot) = o(1/\alpha^2)$, and there is an algorithm $\mathcal{A}$ such that for every $\alpha, \beta \in (0, 1)$, when $\mathcal{A}$ is given private and public samples of sizes $n_{\mathsf{priv}} = n_{\mathsf{priv}}(\alpha, \beta)$, and $n_{\mathsf{pub}} = n_{\mathsf{pub}}(\alpha, \beta)$, it ($\alpha, \beta, 0.1, \mathsf{negl}(n_{\mathsf{priv}})$)-semi-privately learns $\mathcal{H}$.*

Note that in the definition above, the privacy parameters are set as follows: $\epsilon = 0.1$ and $\delta$ is negligible function in the private sample size (and $\delta = 0$ for a *pure* semi-private learner).

The choice of $n_{\mathsf{pub}} = o(1/\alpha^2)$ in the above definition is because taking $\Omega(\mathsf{VC}(\mathcal{H})/\alpha^2)$ public examples suffices to learn the class without any private examples (see [SSBD14]). Thus, the above definition focuses on classes for which there is a non-trivial saving in the number of public examples required for learning. Beimel et al. [BNS13] were the first to propose the notion of *semi-private* learners. Their notion is analogous to a special case of Definition 2.2, which we define next.

**Definition 2.4** (($\alpha, \beta, \epsilon, \delta$)-Semi-Supervised Semi-Private Learner)**.** *The definition is analogous to Definition 2.2 except that the public sample is* unlabeled*. That is, $\mathcal{A}$ is ($\alpha, \beta, \epsilon, \delta$)-SS-SP (semi-supervised semi-private) learner for a class $\mathcal{H}$ with private sample size $n_{\mathsf{priv}}$ and public sample size $n_{\mathsf{pub}}$ if the same conditions in Definition 2.2 hold except that in condition 1, $S_{\mathsf{pub}} \sim \mathcal{D}^{n_{\mathsf{pub}}}$ is replaced with $T_{\mathsf{pub}} \sim \mathcal{D}_{\mathcal{X}}^{n_{\mathsf{pub}}}$, and condition 2 is replaced with "For all $T \in \mathcal{X}^{n_{\mathsf{pub}}}$, $\mathcal{A}(\cdot, T)$ is ($\epsilon, \delta$)-differentially private."*

We define the notion of semi-supervised semi-privately learnable class $\mathcal{H}$ in analogous manner as in Definition 2.3.

**Private learning without public data:** In the standard setting of ($\epsilon, \delta$)-differentially private learning, the learner has no access to public data. We note that this setting can be viewed as a special case of Definitions 2.2 and 2.4 by taking $n_{\mathsf{pub}} = 0$ (i.e., empty public dataset). In such case, we refer to the learner as ($\alpha, \beta, \epsilon, \delta$)-*private learner*. As before, when $\delta = 0$, we call the learner *pure private learner*. The notion of *privately learnable* class $\mathcal{H}$ is defined analogously to Definition 2.3 with $n_{\mathsf{pub}}(\alpha, \beta) = 0$ for all $\alpha, \beta$.

We will use the following lemma due to Beimel et al. [BNS15]:

**Lemma 2.5** (Special case of Theorem 4.16 in [BNS15]). *Any class $\mathcal{H}$ that is privately learnable with respect to all realizable distributions is also privately learnable (i.e., privately learnable in the general agnostic setting).*

The following fact follows from the private boosting technique due to [DRV10]:

**Lemma 2.6** (follows from Theorem 6.1 [DRV10] (the full version)). *For any class $\mathcal{H}$, under the realizability assumption, if there is a $(0.1, 0.1, 0.1)$-pure private learner for $\mathcal{H}$, then $\mathcal{H}$ is privately learnable by a pure private algorithm.*

We note that no analogous statement to the one in Lemma 2.6 is known for approximate private learners. This is because it is not clear how one can scale down the $\delta$ parameter of the boosted learner to $\mathsf{negl}(n)$, as required by the definition of approximate DP learnability; specifically, the boosting result in [DRV10] does not achieve this. On the other hand, the $\epsilon$ parameter of the boosted learner (according to [DRV10, Theorem 6.1]) can be scaled down by taking the input sample of the boosted learner to be large enough, and then apply the algorithm on a random subsample. The same technique would not be sufficient to scale down $\delta$ to $\mathsf{negl}(n)$.

We will also use the following notion of coverings:

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

$$h_1 \Delta h_2(x) \triangleq \mathbf{1}\left(h_1(x) \neq h_2(x)\right) \quad \forall x \in \mathcal{X}$$

Let $\mathcal{G}_{\mathcal{H}_\Delta}$ denote the growth function of $\mathcal{H}_\Delta$; that is, for any number $m$,

$$\mathcal{G}_{\mathcal{H}_\Delta}(m) \triangleq \max_{V:|V|=m} |\Pi_{\mathcal{H}_\Delta}(V)|,$$

where $\Pi_{\mathcal{H}_\Delta}(V)$ is the set of all possible dichotomies that can be generated by $\mathcal{H}_\Delta$ on a set $V$ of size $m$. Note that $\mathcal{G}_{\mathcal{H}_\Delta}(m) \le \left(\frac{e\,m}{d}\right)^{2d}$. This follows from the fact that for any set $V$ of size $m$, we have $|\Pi_{\mathcal{H}_\Delta}(V)| \le |\Pi_{\mathcal{H}}(V)|^2$ since every dichotomy in $\Pi_{\mathcal{H}_\Delta}$ is determined by a pair of dichotomies in $\Pi_{\mathcal{H}}(V)$. Hence, $\mathcal{G}_{\mathcal{H}_\Delta}(m) \le (\mathcal{G}_{\mathcal{H}}(m))^2 \le \left(\frac{e\,m}{d}\right)^{2d}$, where the last inequality follows from Sauer's Lemma [Sau72]. Now, by invoking a uniform convergence argument, we have

$$\mathbb{P}_{T_{\mathsf{pub}} \sim \mathcal{D}_{\mathcal{X}}^{n_{\mathsf{pub}}}} \left[\exists h_1, h_2 \in \mathcal{H}: \ \mathsf{dis}\,(h_1, h_2;\, \mathcal{D}_{\mathcal{X}}) > \alpha \ \text{and} \ \widehat{\mathsf{dis}}\,(h_1, h_2;\, T_{\mathsf{pub}}) = 0\right]$$

$$= \mathbb{P}_{T_{\mathsf{pub}} \sim \mathcal{D}_{\mathcal{X}}^{n_{\mathsf{pub}}}} \left[\exists h \in \mathcal{H}_\Delta: \ \mathsf{dis}\,(h, h_{\mathbf{0}};\, \mathcal{D}_{\mathcal{X}}) > \alpha \ \text{and} \ \widehat{\mathsf{dis}}\,(h, h_{\mathbf{0}};\, T_{\mathsf{pub}}) = 0\right]$$

$$\le 2\mathcal{G}_{\mathcal{H}_\Delta}(2\,n_{\mathsf{pub}})\, e^{-\alpha\, n_{\mathsf{pub}}/4}$$

$$\le 2\left(\frac{2e\,n_{\mathsf{pub}}}{d}\right)^{2d} e^{-\alpha\, n_{\mathsf{pub}}/4}.$$

The bound in the third line is non-trivial; it follows from the so-called double-sample argument which was used by Vapnik and Chervonenkis in their seminal paper [VC15]. The same argument is used in virtually all VC-based uniform convergence bounds (see, e.g., [SSBD14, Sec. 28.3]).

This proves inequality (1) and completes the proof of the lemma.

$\square$

## Proof of the Upper Bound (Theorem 3.1)

First, we note that $\epsilon$-differential privacy of $\mathcal{A}_{\mathsf{SSPP}}$ follows from Lemma 3.2. Thus, it is left to establish the accuracy guarantee of $\mathcal{A}_{\mathsf{SSPP}}$ and the sample complexity bounds on $n_{\mathsf{pub}}$ and $n_{\mathsf{priv}}$. Let

$$h^* \in \arg\min_{h \in \mathcal{H}} \mathsf{err}\,(h;\, \mathcal{D})$$

denote the optimal hypothesis in $\mathcal{H}$. We will show that with probability $\ge 1 - \beta$, the output hypothesis $h_{\mathsf{priv}}$ satisfies $\mathsf{err}\,(h_{\mathsf{priv}};\, \mathcal{D}) \le \mathsf{err}\,(h^*;\, \mathcal{D}) + \alpha$.

First, fix the randomness in the choice of $T_{\mathsf{pub}}$. Let $\widetilde{\mathcal{H}}_{T_{\mathsf{pub}}}$ denote the corresponding realization of the finite class generated in Steps 3-5 of Algorithm 1. Let

$$h^*_{T_{\mathsf{pub}}} \triangleq \arg\min_{h \in \widetilde{\mathcal{H}}_{T_{\mathsf{pub}}}} \mathsf{err}(h;\, \mathcal{D})$$

denote the optimal hypothesis in $\widetilde{\mathcal{H}}_{T_{\mathsf{pub}}}$. Using the result in [KLN$^+$08, Theorem 3.4] for the generic learner based on the exponential mechanism, it follows that a private sample size

$$n_{\mathsf{priv}} = O\left(\left(\log\left(|\widetilde{\mathcal{H}}_{T_{\mathsf{pub}}}|\right) + \log(1/\beta)\right)\max\left(\frac{1}{\alpha^2}, \frac{1}{\epsilon\,\alpha}\right)\right)$$

suffices to ensure that, w.p. $\ge 1 - \beta/2$ (over randomness in $S_{\mathsf{priv}}$ and in the exponential mechanism), we have $\mathsf{err}\,(h_{\mathsf{priv}};\, \mathcal{D}) \le \mathsf{err}\left(\hat{h}_{T_{\mathsf{pub}}};\, \mathcal{D}\right) + \alpha/2$. From the setting of $n_{\mathsf{pub}}$ in the theorem statement together with Sauer's Lemma, it follow that

$$\log\left(|\widetilde{\mathcal{H}}_{T_{\mathsf{pub}}}|\right) \le d\,\log(\frac{e\,n_{\mathsf{pub}}}{d}) \le O\big(d\log(\frac{d\log(1/\alpha) + \log(1/\beta)}{d\alpha})\big)$$

$$= O\left(d\left(\log(1/\alpha) + \log\left(\log(1/\alpha) + \frac{\log(1/\beta)}{d}\right)\right)\right)$$

$$= O\left(d\log(1/\alpha) + d\left(\log\left(\frac{\log(1/\beta)}{d}\right)\right)^+\right)$$

where $(x)^+ \triangleq \max(0, x)$.

Hence,

$$n_{\mathsf{priv}} = O\left(\left(\left(d\log(1/\alpha) + d\left(\log\left(\frac{\log(1/\beta)}{d}\right)\right)\right)^+ + \log(1/\beta)\right) \max\left(\frac{1}{\alpha^2}, \frac{1}{\epsilon\,\alpha}\right)\right)$$

$$= O\left(\left(d\log\left(1/\alpha\right) + \log\left(1/\beta\right)\right)\max\left(\frac{1}{\alpha^2}, \frac{1}{\epsilon\,\alpha}\right)\right)$$

This yields the bound on $n_{\mathsf{priv}}$ as in the theorem statement. Now, by invoking Lemma 3.3, it follows that for the setting of $n_{\mathsf{pub}}$ as in the theorem statement, w.p. $\geq 1 - \beta/2$ over the randomness in $T_{\mathsf{pub}}$, we have $\mathsf{dis}\left(\hat{h}_{T_{\mathsf{pub}}}, h^*; \mathcal{D}_{\mathcal{X}}\right) \leq \alpha/2$. Hence, by the triangle inequality, $\mathsf{err}\left(\hat{h}_{T_{\mathsf{pub}}}; \mathcal{D}\right) - \mathsf{err}\left(h^*; \mathcal{D}\right) \leq \mathsf{dis}\left(\hat{h}_{T_{\mathsf{pub}}}, h^*; \mathcal{D}_{\mathcal{X}}\right) \leq \alpha/2$. This completes the proof of the theorem.

# 4  Lower Bound

In this section we establish that the upper bound on the public sample complexity which was derived in the previous section is nearly tight.

**Theorem 4.1** (Lower bound for classes of infinite Littlestone dimension)**.** *Let $\mathcal{H}$ be any class with an infinite Littlestone dimension (e.g., the class of thresholds over $\mathbb{R}$). Then, any semi-private learner for $\mathcal{H}$ must have public sample of size $n_{\mathsf{pub}} = \Omega(1/\alpha)$, where $\alpha$ is the excess error.*

In the case of pure differentially privacy we get a stronger statement which manifests a dichotomy that applies for every class:

**Theorem 4.2** (Pure private vs. pure semi-private learners)**.** *Every class $\mathcal{H}$ must satisfy exactly one of the following:*

1. *$\mathcal{H}$ is learnable by a pure private learner.*

2. *Any pure semi-private learner for $\mathcal{H}$ must have public sample of size $n_{\mathsf{pub}} = \Omega(1/\alpha)$, where $\alpha$ is the excess error*

**Proof overview.** The crux of the argument is a *public-data-reduction lemma* (Lemma 4.4), which shows how one can reduce the number of public examples at the price of a proportional increase in the excess error. This lemma implies, for example, that if $\mathcal{H}$ can be learned up to an excess error of $\alpha$ with less than $\frac{1}{1000\alpha}$ public examples then it can also be privately learned without any public examples and excess error of at most $< \frac{1}{10}$. Stating contra-positively, if $\mathcal{H}$ can not be privately learned with excess error $< \frac{1}{10}$ then it can not be semi-privately learned up to an excess error of $\alpha$ with less than $\frac{1}{1000\alpha}$ public examples. This yields a lower bound of $\Omega(1/\alpha)$ on the public sample complexity for every class $\mathcal{H}$ which is not privately learnable with constant excess error

One example for such a class is any class with infinite Littlestone dimension (e.g., the class of 1-dimensional thresholds over an infinite domain). This follows from the result in [ALMM18]:

**Theorem 4.3** (Restatement of Corollary 2 in [ALMM18])**.** *Let $\mathcal{H}$ be any class of infinite Littlestone dimension (e.g., the class of thresholds over an infinite domain $\mathcal{X} \subseteq \mathbb{R}$). For any $n \in \mathbb{N}$, given a private sample of size $n$, there is no $\left(\frac{1}{16}, \frac{1}{16}, 0.1, \frac{1}{100\,n^2\log(n)}\right)$-private learner for $\mathcal{H}$ (even in the realizable case).*

A special case of the above result was first proven in [BNSV15], where it was shown that no *proper* private learner can learn thresholds over an infinite domain. A proper learner is bound to output a hypothesis from the given class. Our definitions in this paper for private and semi-private learners do not make this restriction on the learner; that is, the learners in those definitions can be *non-proper*, i.e., they are allowed to output a binary hypothesis that is not necessarily in the given class $\mathcal{H}$.

**Remark 1.** *The aforementioned reduction we use for the lower bound holds even when the public sample is* labeled. *This makes the lower bound stronger since it holds even in the fully supervised setting of semi-private learning described in Definition 2.2. We also note that this reduction holds for both* pure *and* approximate *private/semi-private learners.*

We now formally state and prove the reduction outlined above.

**Lemma 4.4** (Public data reduction lemma). *Let $0 < \alpha \leq 1/100$, $\epsilon > 0$, $\delta \geq 0$. Suppose there is an $(\alpha, \frac{1}{18}, \epsilon, \delta)$-agnostic semi-private learner for a hypothesis class $\mathcal{H}$ with private sample size $n_{\mathsf{priv}}$ and public sample size $n_{\mathsf{pub}}$. Then, there is a $\left(100\, n_{\mathsf{pub}}\, \alpha,\, \frac{1}{16}, \epsilon, \delta\right)$-private learner that learns any distribution realizable by $\mathcal{H}$ with input sample size $\left\lceil \frac{n_{\mathsf{priv}}}{10\, n_{\mathsf{pub}}} \right\rceil$.*

*Proof.* Let $\mathcal{A}$ denote the assumed agnostic-case semi-private learner for $\mathcal{H}$ with input private sample of size $n_{\mathsf{priv}}$ and input public sample of size $n_{\mathsf{pub}}$. Using $\mathcal{A}$, we construct a realizable-case private learner for $\mathcal{H}$, which we denote by $\mathcal{B}$. The description of $\mathcal{B}$ appears in Algorithm 2.

---

**Algorithm 2** Description of the private learner $\mathcal{B}$:

---

**Input:** Private sample $\tilde{S} = (\tilde{z}_1, \ldots, \tilde{z}_{\tilde{n}})$ of size $\tilde{n} = \lceil n_{\mathsf{priv}}/(10 \cdot n_{\mathsf{pub}}) \rceil$.
1: Pick a fixed (dummy) distribution $\mathcal{D}_0$ over $\mathcal{Z} = \mathcal{X} \times \{0,1\}$ where the label $y \in \{0,1\}$ is drawn uniformly at random from $\{0,1\}$ independently from $x \in \mathcal{X}$.
2: Set $p = 1/(100 \cdot n_{\mathsf{pub}})$.
3: Using $\tilde{S}$ and $\mathcal{D}_0$, construct samples $S_{\mathsf{priv}}$, $S_{\mathsf{pub}}$ using procedures $\mathsf{PrivSamp}(\tilde{S}, \mathcal{D}_0, p, n_{\mathsf{priv}})$ and $\mathsf{PubSamp}(\tilde{S}, \mathcal{D}_0, n_{\mathsf{pub}})$ given by Algorithms 3 and 4 below.
4: Return $\tilde{h} = \mathcal{A}(S_{\mathsf{priv}}, S_{\mathsf{pub}})$.

---

**Algorithm 3** Private Sample Generator $\mathsf{PrivSamp}$:

---

**Input:** Sample $\tilde{S} = (\tilde{z}_1, \ldots, \tilde{z}_{\tilde{n}})$, Distribution $\mathcal{D}_0$, parameter $p$, sample size $n_{\mathsf{priv}}$.
1: $i := 1$
2: **while** $\tilde{S} \neq \emptyset$ **and** $i \leq n_{\mathsf{priv}}$: **do**
3:     Sample $b_i \sim \mathsf{Ber}(p)$ (independently for each $i$), where $\mathsf{Ber}(p)$ is Bernoulli distribution with mean $p$.
4:     **if** $b_i = 1$: **then**
5:         Set $z_i^{\mathsf{prv}}$ to be the next element in $\tilde{S}$, i.e., $z_i^{\mathsf{prv}} = \tilde{z}_{j_i}$, where $j_i = \sum_{k=1}^{i} b_k$.
6:         Remove this element from $\tilde{S}$: $\tilde{S} \leftarrow \tilde{S} \setminus \tilde{z}_{j_i}$.
7:     **else**
8:         Set $z_i^{\mathsf{prv}} = z_i^0$, where $z_i^0$ is a fresh independent example from the "dummy" distribution $\mathcal{D}_0$.
9:     $i \leftarrow i + 1$
10: **return** $S_{\mathsf{priv}} = (z_1^{\mathsf{prv}}, \ldots, z_{n_{\mathsf{priv}}}^{\mathsf{prv}})$.

---

**Algorithm 4** Public Sample Generator $\mathsf{PubSamp}$:

---

**Input:** Sample $\tilde{S} = (\tilde{z}_1, \ldots, \tilde{z}_{\tilde{n}})$, Distribution $\mathcal{D}_0$, sample size $n_{\mathsf{pub}}$.
1: **for** $i = 1, \ldots, n_{\mathsf{pub}}$ : **do**
2:     Set $z_i^{\mathsf{pub}} = z_i^0$ where $z_i^0$ is a fresh independent example from $\mathcal{D}_0$.
3: **return** $S_{\mathsf{pub}} = (z_1^{\mathsf{pub}}, \ldots, z_{n_{\mathsf{pub}}}^{\mathsf{pub}})$

---

The following two claims about $\mathcal{B}$ establish its privacy and accuracy guarantees.

**Claim 4.5** (Privacy guarantee of $\mathcal{B}$). $\mathcal{B}$ *is $(\epsilon, \delta)$-differentially private*

This follows directly from the fact that for any realization of $S_{\mathsf{pub}}$, $\mathcal{A}(\cdot, S_{\mathsf{pub}})$ is $(\epsilon, \delta)$-differentially private, the fact that $S_{\mathsf{pub}}$ does not contain any points from $\tilde{S}$, and the fact that each point in $\tilde{S}$ appears at most once in $S_{\mathsf{priv}}$.

Thus, it remains to show that

**Claim 4.6** (Accuracy guarantee of $\mathcal{B}$). *Let $\mathcal{D}$ be any distribution over $\mathcal{Z}$ that is realizable by $\mathcal{H}$. Suppose $\tilde{S} \sim \mathcal{D}^{\tilde{n}}$. Then, except with probability at most $1/16$ (over the choice of $\tilde{S}$ and internal randomness in $\mathcal{B}$), the output hypothesis $\tilde{h}$ satisfies:* $\mathrm{err}(\tilde{h}; \mathcal{D}) \leq 100\, n_{\mathsf{pub}}\, \alpha$.

Let $\mathcal{D}_{(p)}$ denote the mixture distribution $p \cdot \mathcal{D} + (1-p) \cdot \mathcal{D}_0$ (recall the definition of $p$ from Algorithm 2). To prove Claim 4.6, we first show that both $S_{\mathsf{priv}}$ and $S_{\mathsf{pub}}$ can be viewed as being sampled from $\mathcal{D}_{(p)}$ with almost no impact on the analysis. Then, using the fact that $\mathcal{A}$ learns $\mathcal{H}$ with respect to $\mathcal{D}_{(p)}$, the claim will follow.

First, note that since $\tilde{n} = 10 \cdot p \cdot n_{\mathsf{priv}}$, then by Chernoff's bound, except with probability $< 0.01$, Algorithm 3 exits the **WHILE** loop with $i = n_{\mathsf{priv}}$. Thus, except with probability $< 0.01$, we have

$$|S_{\mathsf{priv}}| = n_{\mathsf{priv}}, \quad \text{hence,} \quad S_{\mathsf{priv}} \sim \mathcal{D}_{(p)}^{n_{\mathsf{priv}}}. \tag{2}$$

As for $S_{\mathsf{pub}}$, note that $S_{\mathsf{pub}} = (z_1^0, \ldots, z_{n_{\mathsf{pub}}}^0) \sim \mathcal{D}_0^{n_{\mathsf{pub}}}$, and therefore we can not use the same argument we used with $S_{\mathsf{priv}}$. Instead, we will show that $\mathcal{D}_0^{n_{\mathsf{pub}}}$ is close in total variation to $\mathcal{D}_{(p)}^{n_{\mathsf{pub}}}$. Let $\widehat{S}_{\mathsf{pub}} = (\hat{z}_1, \ldots, \hat{z}_{n_{\mathsf{pub}}})$ be i.i.d. sequence generated as follows: for each $i \in [n_{\mathsf{pub}}]$, $\hat{z}_i = b_i\, v_i + (1 - b_i)\, z_i^0$, where $(b_1, \ldots, b_{n_{\mathsf{pub}}}) \sim (\mathsf{Ber}(p))^{n_{\mathsf{pub}}}$, and $(v_1, \ldots, v_n) \sim \mathcal{D}^{n_{\mathsf{pub}}}$. It is clear that $\widehat{S}_{\mathsf{pub}} \sim \mathcal{D}_{(p)}^{n_{\mathsf{pub}}}$. Moreover, observe that

$$\mathbb{P}\left[\widehat{S}_{\mathsf{pub}} = S_{\mathsf{pub}}\right] \geq \mathbb{P}\left[b_i = 0 \ \forall\, i \in [n_{\mathsf{pub}}]\right] = \left(1 - \frac{1}{100\, n_{\mathsf{pub}}}\right)^{n_{\mathsf{pub}}} \geq 0.99$$

This implies that the total variation between $\widehat{S}_{\mathsf{pub}}$ and $S_{\mathsf{pub}}$ is at most $0.01$. In particular, the probability of any event w.r.t. the distribution of $\widehat{S}_{\mathsf{pub}}$ is at most $0.01$ far from the probability of the same event w.r.t. the distribution of $S_{\mathsf{pub}}$. Hence,

$$\mathbb{P}_{S_{\mathsf{priv}}, S_{\mathsf{pub}}, \mathcal{A}} \left[ \mathrm{err}\left(\mathcal{A}(S_{\mathsf{priv}}, S_{\mathsf{pub}}); \mathcal{D}_{(p)}\right) - \min_{h \in \mathcal{H}} \mathrm{err}(h; \mathcal{D}_{(p)}) > \alpha \right]$$
$$- \mathbb{P}_{S_{\mathsf{priv}}, \widehat{S}_{\mathsf{pub}}, \mathcal{A}} \left[ \mathrm{err}\left(\mathcal{A}(S_{\mathsf{priv}}, S_{\mathsf{pub}}); \mathcal{D}_{(p)}\right) - \min_{h \in \mathcal{H}} \mathrm{err}(h; \mathcal{D}_{(p)}) > \alpha \right] \leq 0.01 \tag{3}$$

Now, from (2) and the premise that $\mathcal{A}$ is agnostic semi-private learner, we have

$$\mathbb{P}_{S_{\mathsf{priv}}, \widehat{S}_{\mathsf{pub}}, \mathcal{A}} \left[ \mathrm{err}\left(\mathcal{A}(S_{\mathsf{priv}}, S_{\mathsf{pub}}); \mathcal{D}_{(p)}\right) - \min_{h \in \mathcal{H}} \mathrm{err}(h; \mathcal{D}_{(p)}) > \alpha \right] \leq \frac{1}{17}$$

Hence, using (3), we conclude that except with probability $< 1/16$,

$$\mathrm{err}\left(\mathcal{A}(S_{\mathsf{priv}}, S_{\mathsf{pub}}); \mathcal{D}_{(p)}\right) - \min_{h \in \mathcal{H}} \mathrm{err}(h; \mathcal{D}_{(p)}) \leq \alpha. \tag{4}$$

Note that for any hypothesis $h$,

$$\mathrm{err}(h; \mathcal{D}_{(p)}) = p \cdot \mathrm{err}(h; \mathcal{D}) + (1-p) \cdot \mathrm{err}(h; \mathcal{D}_0) = p \cdot \mathrm{err}(h; \mathcal{D}) + \frac{1}{2}(1-p),$$

where the last equality follows from the fact that the labels generated by $\mathcal{D}_0$ are completely noisy (uniformly random labels). Hence, we have $\arg\min_{h \in \mathcal{H}} \mathrm{err}(h; \mathcal{D}_{(p)}) = \arg\min_{h \in \mathcal{H}} \mathrm{err}(h; \mathcal{D})$. That is, the optimal hypothesis with respect to the realizable distribution $\mathcal{D}$ is also optimal with respect to the mixture distribution $\mathcal{D}_{(p)}$. Let $h^* \in \mathcal{H}$ denote such hypothesis. Note that $\mathrm{err}(h^*; \mathcal{D}) = 0$ and $\mathrm{err}(h^*; \mathcal{D}_{(p)}) = \frac{1}{2}(1-p)$. These observations together with (4) imply that except with probability $< 1/16$, we have

$$\alpha \geq p \cdot \mathrm{err}\left(\mathcal{A}(S_{\mathsf{priv}}, S_{\mathsf{pub}}); \mathcal{D}\right)$$

Hence, $\mathrm{err}\left(\mathcal{B}(\tilde{S}); \mathcal{D}\right) = \mathrm{err}\left(\mathcal{A}(S_{\mathsf{priv}}, S_{\mathsf{pub}}); \mathcal{D}\right) \leq 100 \cdot n_{\mathsf{pub}} \cdot \alpha$. This completes the proof.

$\square$

With Lemma 4.4, we are now ready to prove the main results for this section:

**Proof of Theorem 4.1**

*Proof.* Suppose $\mathcal{A}$ is a semi-private learner for $\mathcal{H}$ with sample complexities $n_{\text{priv}}, n_{\text{pub}}$. In particular, given $n_{\text{priv}}(\alpha, \frac{1}{18}), n_{\text{pub}}(\alpha, \frac{1}{18})$ private and public examples, $\mathcal{A}$ is $(\alpha, \frac{1}{18}, 0.1, \frac{1}{100\, n_{\text{priv}}^2 \log(n_{\text{priv}})})$-semi-private learner for $\mathcal{H}$. Hence, by Lemma 4.4, there is $(100 n_{\text{pub}}\alpha, \frac{1}{16}, 0.1, \frac{1}{100\, n_{\text{priv}}^2 \log(n_{\text{priv}})})$-private learner for $\mathcal{H}$. Thus, Theorem 4.3 implies that $100 n_{\text{pub}}\alpha > \frac{1}{16}$ and hence that $n_{\text{pub}} > \frac{1}{1600\, \alpha}$ as required. $\square$

**Proof of Theorem 4.2**

*Proof.* First, if $\mathcal{H}$ is learnable by a pure private learner, then trivially the second condition cannot hold since $\mathcal{H}$ can be learned without any public examples. Now, suppose that the first item does *not* hold. Note that by Lemma 2.5, this implies that there is *no* pure private learner for $\mathcal{H}$ with respect to realizable distributions. By Lemma 2.6, this in turn implies that there is *no* $\left(\frac{1}{16}, \frac{1}{16}, 0.1\right)$-pure private learner for $\mathcal{H}$ with respect to realizable distributions. Now, suppose $\mathcal{A}$ is a pure semi-private learner $\mathcal{A}$ for $\mathcal{H}$. Then, this implies that for any $\alpha > 0$, $\mathcal{A}$ is an $\left(\alpha, \frac{1}{18}, 0.1\right)$-pure semi-private learner for $\mathcal{H}$ with sample complexities $n_{\text{priv}}(\alpha, \frac{1}{18}), n_{\text{pub}}(\alpha, \frac{1}{18})$. Hence, by Lemma 4.4, there is a $\left(100\, n_{\text{pub}}\, \alpha, \frac{1}{16}, 0.1\right)$-pure private learner for $\mathcal{H}$ w.r.t. realizable distributions. This together with the earlier conclusion implies that $100\, n_{\text{pub}}\, \alpha > \frac{1}{16}$, and therefore that $n_{\text{pub}} > \frac{1}{1600\, \alpha}$, which shows that the condition in the second item holds. $\square$

## Footnotes

[1]The Littlestone dimension is a combinatorial parameter that arises in online learning [Lit87, BPS09].

[2]The exponential mechanism is a basic algorithmic technique in DP [MT07].