[Reviews · NeurIPS 2019]

Reviewer 1



The authors studied the sample complexity of semi-private learning a VC class. The paper is well-written and clear. The first main result is that any VC class can be (agnostically) learned with VC(H)/alpha^2 private data and VC(H)/alpha public data. The algorithm is the same as [BNS13], which studied the realizable case, but the analysis in this paper is novel. The second main result is that when H has infinite littlestone dimension, then any private learner must have at least 1/alpha public sample complexity, therefore the upper bound is nearly tight. The proof is based on a result from a recent STOC paper and a new "public data reduction lemma". As a consequence of the lemma, the authors showed a dichotomy of pure semi-private learning. Overall, this paper provides solid and near-optimal theoretical results on semi-private learning. I suggest acceptance.

Reviewer 2



Originality: The methods used in this work are simple and for the most part standard in the literature of learning theory. The idea of learning \alpha-coverings is nice, and to the best of my knowledge, new. Quality: The statements of the Theorems and Lemma are careful and make sense. Every proof I checked seems correct. I would prefer though if the authors mention more clearly that their results are significant only in the agnostic setting; in the realizable setting I think VC(H)/\alpha sample size is sufficient and the private sample size can be ignored. Clarity: I consider the paper very well written and the exposure sufficiently clear. Significance: The paper seems significant and I can see future research to cite this work and be using their techniques.

Reviewer 3



[Upper bound] Every hypothesis class H can be learned up to excess error α by a pure semi- private algorithm whose private sample complexity is (roughly) VC(H)/α^2 and public sample complexity is (roughly) VC(H)/α. For agnostic learning, VC(H)/α2 examples are necessary even without privacy constraints. The idea of algorithm is to use the public data to construct a finite class H′ that forms a “good approximation” of the original class H, then reduce the problem to DP learning of a finite class. It can be captured via the notion of α-covering. [Lower bound] Assume H has an infinite Littlestone dimension. Then, any approximate semi- private learner for H must have public sample complexity Ω(1/α). The lower bounds boil down to a public-data-reduction lemma which shows that given a semi-private learner whose public sample complexity is << 1/α, transforms it to a completely. Overall, this paper is clear, written well, and has a good contribution. From my view, it is around the conference threshold.

[Author Response · NeurIPS 2019]

We thank all the reviewers for their comments.

**Responses to Reviewer-2's comments:**

*"... would prefer if the authors mention more clearly that their results are significant only in the agnostic setting.... "*

Indeed. We mentioned that our work focuses on the agnostic setting in several places (including the abstract and the
introduction), but we will elaborate more on this point as suggested by Reviewer 2.

*" Is there [..] intuitive way to explain why there is such a discontinuity at public sample size $1/\alpha$?"*

Here is one way to think about this: this kind of sharp transition is a by-product of the fact that the definition of
PAC learnability is a worst-case (min-max style) definition. Similar discontinuities are also exhibited by standard
(non-private) PAC sample complexity bounds: for example, a class is either learnable with $O(VC(H)/\alpha^2)$ examples,
or it is not learnable at all (if $VC(H) = \infty$).

*"Is there any way the lower bound on the public sample size to become $VC(H)/\alpha$ instead of $1/\alpha$? ... I would suggest*
*the authors to mention whether this is a hard next research step or not."*

This is a very good question. Although it is natural to think that the upper bound should be tight, it is not immediately
obvious, at least for general VC classes, how to involve this factor of $VC(H)$ in the lower bound. We believe this to be
an interesting research question.

*"what does the term $negl(n_{priv})$ mean in Definition 2.3?"*

This means it is a negligible function of $n_{priv}$. The function $negl(.)$ is formally defined earlier in the first paragraph of
Section 2.

*"In Algorithm 1, step 5: By "add to $\tilde{H}$ arbitrary $h$.." do you mean "add to $\tilde{H}$ every $h$.." or "add to $\tilde{H}$ one $h$ arbitrarily*
*chosen.." ? I suspect the former but it is not clear."*

It is the latter. To construct the $\alpha$-cover, one only needs one representative hypothesis (chosen arbitrarily) for each
dichotomy. We will rephrase this step to make it entirely clear.

**Response to Reviewer-3's comments:**

*"For the lower bound, it seems not very complete. Authors show that if a concept can't be pure privately learned, then any*
*semi-private learner must have $\Omega(1/\alpha)$ public samples. So there is a problem, does a non-trivial semi-private learner*
*for this concept always exist? Non-trivial means that the learner doesn't learn only from the public data, otherwise,*
*there is no privacy issue in this learning. If a concept can't be semi-privately learned nontrivially, then the lower bound*
*has no sense. Recall that they show an algorithm for semi-privately learning a concept with finite VC dimension, then*
*whether there is a semi-private algorithm for infinite VC dimension, this is not clear."*

We have not been able to understand the comment. If the VC-dimension is infinite, then learning is impossible, even
ignoring any privacy issues. On the other hand, if the Littlestone dimension is infinite, then private leaning is impossible.
Thus the lower bound is interesting mainly when the VC-dimension is finite and the Littlestone dimension is infinite. In
this case our positive result shows that a non-trivial semi-private learner always exists, indeed the learner needs only
$VC/\alpha$ public examples and hence does not learn only from the public examples as altogether $VC/\alpha^2$ examples are
needed for learning in the general agnostic setting, which is the setting we focus on in this work. Our lower bound
shows that the dependence on $\alpha$ in the number of public examples for this non-trivial semi-private learner is tight.

*"There are some typos and expressions can be fixed: Line 82, it should be $VC/\alpha$, rather than $VC/\alpha^2$"*

This is not a typo. What we are saying here is that constructing an $\alpha$-cover using $VC/\alpha^2$ examples is rather
straightforward using standard uniform convergence arguments. Hence, a construction (like ours) that involves only
$VC/\alpha$ public examples is non-trivial.

*"Line 270: [This implies that the total variation between $\hat{S}$ and $S$ is at most 0.01.] This sentence is confusing. The above*
*inequality means that the probability of $\hat{S} \neq S$ is at most $0.01$. How does the total variation mean here?"*

This follows from the sequence of steps before that line. We are happy to elaborate and will include this clarification in
the paper. First, note that the distribution of the examples in $\hat{S}_{pub}$ is a mixture of two distributions $b \cdot D + (1-b) \cdot D_0$,
where $D$ is the original distribution (realizable by $H$), and $D_0$ is the distribution of the examples in $S_{pub}$. Second, note
that the probability that $\hat{S}_{pub} \neq S_{pub}$ is an upper bound on the measure attributed to the first component of the mixture
distribution of $\hat{S}_{pub}$ (i.e., the component from $D$). Hence, it follows that the total variation between the distribution of
$\hat{S}_{pub}$ (induced by the mixture) and the distribution of $S_{pub}$ (induced by $D_0$) is upper bounded by the aforementioned
quantity.

[Meta-Review · NeurIPS 2019]

The three reviewers made positive assessment on this work. My view on the significance of the paper is highly positive since it develops a solution to private learning which answers a known limitation for the case of VC classes. I recommend the paper to be accepted as a poster presentation.